# How Do Care Partners of People with Rare Dementia Use Language in Online Peer Support Groups? A Quantitative Text Analysis Study

**DOI:** 10.3390/healthcare12030313

**Published:** 2024-01-25

**Authors:** Oliver S. Hayes, Celine El Baou, Chris J. D. Hardy, Paul M. Camic, Emilie V. Brotherhood, Emma Harding, Sebastian J. Crutch

**Affiliations:** 1Dementia Research Centre, UCL Queen Square Institute of Neurology, UCL, London WC1N 3AR, UKp.camic@ucl.ac.uk (P.M.C.); e.brotherhood@ucl.ac.uk (E.V.B.); s.crutch@ucl.ac.uk (S.J.C.); 2Adapt Lab, Research Department of Clinical, Educational and Health Psychology, UCL, London WC1E 7HB, UK

**Keywords:** dementia, text analysis, rare dementia, LIWC, support group, social comparison theory

## Abstract

We used quantitative text analysis to examine conversations in a series of online support groups attended by care partners of people living with rare dementias (PLWRD). We used transcripts of 14 sessions (>100,000 words) to explore patterns of communication in trained facilitators’ (*n* = 2) and participants’ (*n* = 11) speech and to investigate the impact of session agenda on language use. We investigated the features of their communication via Poisson regression and a clustering algorithm. We also compared their speech with a natural speech corpus. We found that differences to natural speech emerged, notably in emotional tone (d = −3.2, *p* < 0.001) and cognitive processes (d = 2.8, *p* < 0.001). We observed further differences between facilitators and participants and between sessions based on agenda. The clustering algorithm categorised participants’ contributions into three groups: sharing experience, self-reflection, and group processes. We discuss the findings in the context of Social Comparison Theory. We argue that dedicated online spaces have a positive impact on care partners in combatting isolation and stress via affiliation with peers. We then discuss the linguistic mechanisms by which social support was experienced in the group. The present paper has implications for any services seeking insight into how peer support is designed, delivered, and experienced by participants.

## 1. Introduction

### 1.1. Rare Disease Support Options

Online interventions can facilitate participation in groups where factors such as cost, care commitments or travel distance might prevent in-person attendance. Even in cases where support is locally available, care partners’ caregiving responsibilities can prohibit access [1,2]. Care partners, in the present paper, will refer to people who provide support and assistance to an individual with a health condition, though we acknowledge that the individual affected by the condition should be considered an equal partner in their own care [3].

The relative utility of online support groups compared with face-to-face interventions is disputed, but studies have shown both forum-style opportunities (e.g., [4]) and videoconference groups (e.g., [5]) to have positive effects. Findings extend to populations affected by rare diseases such as Cystic Fibrosis [6] and Duchenne’s Muscular Dystrophy [7]. The potential advantages of online intervention are amplified for those affected by rare diseases (conditions affecting fewer than 1 in 2000 people [8]), where targeted support is seldom locally available.

Another population where this is relevant is people living with rare dementias (PLWRD)—by which we mean atypical (e.g., non-memory-led), young-onset (symptoms/diagnosis before age 65) and directly inherited forms of dementia. Care partners of PLWRD experience the access difficulties outlined above. The COVID-19 pandemic heightened the often-reported sensation of isolation in PLWRD and their care partners [9] and emphasised geographical isolation via the implementation of lockdowns and increased hesitancy for travel. Whilst exacerbating the support needs of PLWRD and their care partners [10], the pandemic greatly restricted previously available support. Existing in-person support either ceased [11] or was adapted for online implementation, making the effective implementation of online support paramount.

### 1.2. Quantitative Text Analysis in Psychology

In the present study, we examine multicomponent (peers and professionals) support groups delivered via videoconference for care partners of PLWRD in response to the pandemic. Standardised outcome measures were not collected within these groups as they were organised as a rapid response rather than an experimental intervention. Accordingly, the current study uses transcripts of the group sessions to perform computational linguistic analyses to gain insight into the groups’ mechanisms.

Speech contains multiple levels of information above the word level. Prosody (patterns of stress and intonation) can communicate information about the speaker’s focus of attention and mood [12]. The ways words combine both in meaning (semantically) and form (morphosyntactically) convert constituent parts to new wholes [13] at the phrase and sentence level. The wider context in which the conversation takes place can affect the inferred and implied meanings of the sentences or whole utterances (pragmatic context) [14]. Nevertheless, programmes that are insensitive to all data above the word level can glean a great variety of insight from language data.

The quantitative text analysis software Linguistic Inquiry and Word Count (LIWC2015 version 1.6.0) [15] is uniquely placed to inform on a speaker’s psychological and social states. Developed by researchers interested in social, clinical, health and cognitive psychology [16], LIWC’s linguistic and psychological dimensions offer researchers opportunities to examine both the ‘how’ and the ‘what’ of the participant’s attention and cognition [17]. LIWC has been used extensively in mental health research, for example in investigating thought patterns in the context of grief [18] and in identifying differences in language use when describing previously disclosed and undisclosed events [19]. Researchers have also used LIWC to assess and deepen understanding of the efficacy of support groups; in training machine learning models to identify emotional and informational support in online support groups [20], and in directly identifying how adolescents and young adults with cancer talk about their needs in both face-to-face and online support groups [21].

### 1.3. Isolation in Rare Disease

Davison et al. state that “the experience of illness is a profoundly social one” [22] (p. 205). Illness can precipitate isolation while heightening the need for social interaction. Social comparison theory [23] proposes that prosocial behaviour at the individual level is driven by a desire to hold accurate information about one’s environment and to avoid feeling abnormal or ill-fitting. This framework is especially pertinent in the context of health [22] where anxiety is heightened, and information is often complexly delivered and inconclusive.

Isolation precipitated by the responsibilities of caregiving is a common stressor for care partners. Even care partners of individuals with relatively well-known diseases are often affected. One non-rare dementia study found that 44% of care partners reported moderate loneliness and 18% reported severe loneliness [24].

In the context of rare disease, care partners can find themselves in a less ‘normal’ and less ‘informed’ healthcare setting than those in the study described above (e.g., [25]). For care partners of PLWRD, pathways for support are often confusing. Journeys for diagnoses can be long and arduous [26], and post-diagnostic support is habitually poorly attended [27], suggesting that affected persons may often find available support to be unhelpful. Reviews have highlighted that many affected by rare disease can benefit from specialised psychosocial support to combat stress, burden, and isolation [28,29].

Support groups can form part of the answer for increasing feelings of normalcy, and in elevating the quality of information for individuals in many health contexts. In the context of dementia, the value of support groups for care partners is well-attested, with findings including post-support reductions in depression and perceived burden, and improvements to self-esteem, well-being, and quality of life [30]. Research focusing on participants affected by rare forms of dementia is sparse but online support groups have been shown to be effective in fostering a sense of community [5].

### 1.4. Social Comparison Theory Applied to Health

A recent meta-analysis of the last 60 years of work in the field of Social Comparison Theory [31] offers two important findings for the present context. Firstly, the opportunity for lateral comparison lessens upward comparison (to someone of a higher perceived status). Lateral comparison allows for feeling ‘in the same boat’ as those around you (e.g., [32]), a motivation that competes with the desire for upward comparison. Secondly, while evaluations of contrast are the predominant outcome of comparison, assimilation, can be made more likely in the right contexts.

Care partners of PLWRD often find themselves with a greater grasp of the condition than non-specialised healthcare professionals (e.g., in FTD [33,34]). This situation therefore encourages downward comparison. Our default comparison is upwards, to those perceived to be better or more knowledgeable, and we seek opportunities to do so [31]. It follows that care partners of PLWRD would therefore relish opportunities to be among specialist professionals and other experts by experience (care partners), including other care partners who support PLWRD at more advanced points of disease progression. Care partners finding themselves in such contexts may well find some members to be less knowledgeable than themselves, but fewer members of the environment will fit this evaluation than in non-specialised contexts. The specialised, informed contexts also provide for lateral comparison with others perceived as having equal experience and knowledge levels. Upwards and lateral comparison are rarely facilitated for care partners of PLWRD but support groups with other experts can facilitate both and a well-facilitated multicomponent support group may provide the right context for lateral comparison and assimilation specifically.

### 1.5. Study Aims and Objectives

The present work explores the mechanisms by which support groups invite comparison and affiliation for care partners of PLWRD who often feel isolated and dissimilar to their peers in their wider social contexts. This is achieved by examining the linguistic features of their, and their facilitators’, contributions within a support group context.

Research questions:How does language use in the support groups differ from natural speech?Which linguistic features distinguish facilitator and participant speech?Does the presence and nature of a session agenda direct the language use and thereby modes for social comparison of participants and facilitators?

The efficacy of the groups is inferred from the language use, as are mechanisms by which the groups provide opportunity for upward and lateral comparison. As an exploratory piece of work, the current work generates hypotheses pertaining to the mechanisms of social support in support groups to be considered in future work. Methods reflect this focus with each method partially addressing one or more research questions.

## 2. Methods

### 2.1. Ethical Considerations

The University College London (UCL) Ethics Committee (ref: 8545/004: Rare Dementia Support (RDS) Impact Study) gave ethical approval for the current work. All participants gave informed consent, via a phone or videocall [35].

### 2.2. Recruitment

Rare Dementia Support members (individuals who have given permission to be contacted about research opportunities) received correspondence via email or post with the dates of the support groups, a brief description of the agendas of each support group, and an explanation about the intent to record the meetings for research purposes. Members were made aware that participation was voluntary, and their choice would not affect their membership status. Recruitment closed when each group reached capacity (12 attendees). Participants expressing interest had a short appointment with a Rare Dementia Support researcher both to confirm eligibility and to provide the opportunity to discuss the study and ask further questions.

Purposive sampling took place to achieve a sample with people of differing experiences supporting loved ones with different rare dementias.

### 2.3. Participants

11 Rare Dementia Support members attended the present support group series. At the groups’ onset, all participants (F = 7, M = 4) were care partners for a PLWRD (4 posterior cortical atrophy, 3 frontotemporal dementia, 2 primary progressive aphasia, 1 familial Alzheimer’s disease, and 1 dementia with Lewy bodies), although 1 person later became bereaved. The care partners were recruited from across the UK, in a mixture of urban and rural settings. They were mixed in terms of age (43–71 range) and relationship to the PLWRD to ensure the sample represented a diversity of experiences to increase the richness of the perspectives captured.

Inclusion criteria required participants to be over 18 years old and to have the capacity to understand, retain, assess, and communicate the information required to make the decision to participate in the study. Participants were also required to have a digital device (phone, tablet, or computer) and an internet connection.

The support groups took place virtually via the GoToMeeting (LogMeIn Inc., Boston, MA, USA) video conferencing platform. In accordance with the study procedures outlined in the Participant Information Sheet, the support groups were recorded from the video conferencing platform GoToMeeting with the consent of participants. This platform provides end-to-end security measures preventing active and passive attacks.

### 2.4. Data Pre-Processing of Session Files

Collected data were downloaded from the GoToMeeting portals to secure UCL servers before being uploaded to the UCL Data Safe Haven (certified to the ISO27001 information security standard and conforms to the NHS Digital Data Security and Protection Toolkit) [36].

The recorded data were then transcribed by a third-party transcription service, UK Transcription in accordance with UCL guidance.

### 2.5. Text Corpus

The study’s data source was a text corpus of transcripts of 14 support group discussions centring around independence and identity [37]. Rare Dementia Support set these support groups up as part of their response to growing support demands [10,11,38] caused by the COVID-19 pandemic. The support group sessions in the current corpus were recorded between July 2020 and February 2021.

Two groups, Group 1 (F = 2, M = 3) and Group 2 (F = 5, M = 1) completed four semi-structured sessions separately. These sessions were facilitated by a mental health professional with extensive knowledge of PLWRD care partners (authors S.J.C. and E.H.). Meeting fortnightly, the groups facilitated discussions around four pre-set agendas. All participants were then invited to a single continuing group which met once a month in unstructured sessions (with no agenda). The facilitators’ role in follow-on sessions was reduced; the focus was more on peer support. Detailed information about the recruitment and recording procedure is available in the Appendix A.

Session transcripts were cleaned of personal identifiers creating a pseudonymised dataset before being converted into text files. These were imported into R [39], (a programming language specifically designed for statistical analyses), using a novel script, including one row per contribution (uninterrupted speech from one person delimited by speech from another).

The dataset contained 1320 contributions made up of 129,616 words. Qualifier variables for each contribution were added to the dataset, including session type (semi-structured, unstructured), attendee role (facilitator, participant), and group (Group 1, Group 2, and “Continuing” for unstructured sessions). A summary of the data available is presented in Table 1.

Text analysis was undertaken using LIWC2015 v1.6.0 [13]. LIWC reads text files one word at a time and does not process context (e.g., negation). For each word, LIWC’s dictionary file is searched; if the target word matches a dictionary word, appropriate category scale(s) for that word are incremented [40]. LIWC’s sophisticated stemming system allows the identification of affixed words as having the same core meaning. Words can belong to multiple categories: for example, ‘cried’ fits into the categories of sadness, negative emotion, affective processes, verbs, past focus [40] and more.

### 2.6. Variable Selection

Several LIWC variables were considered for analysis. Criteria for inclusion were either driven by previous literature, or internally driven (considered insightful for current research questions). The list of selected Summary, linguistic, and psychological variables is available in Table 2, Table 3 and Table 4, respectively.

### 2.7. Statistical Analysis

#### 2.7.1. Descriptive Comparisons to LIWC-Defined Natural Speech Summaries

LIWC’s authors provide a series of scored linguistic profiles obtained from corpora of 6 types of texts (blogs, expressive writing, novels, natural speech, New York Times articles, Twitter) [40]. The natural speech corpus includes transcripts from a range of everyday life conversations. To mirror the computation of the natural speech scores provided by LIWC, we aggregated our contribution-level variable scores into average session-level scores, overall (Appendix A), by role (facilitator, participant), and by session type (semi-structured (Appendix A), unstructured (Appendix A)). Scores were then compared to the natural speech corpus scores using one-sample *t*-tests and effect sizes indicated by Cohen’s d [49].

#### 2.7.2. Change in Language Use across Sessions: Mixed Effects Poisson Modelling

To obtain a more granular understanding of language use across sessions and individuals, we segmented the data into participants’ contributions per previous literature [21,50]. LIWC generates a word count per inputted chunk and percentage scores for each of its variables. This combination allowed for simple calculation of word count for each LIWC variable for each contribution.

Mixed effect Poisson models [51] were fitted to estimate the rate of occurrence of each variable per contribution and examine changes, as per previous research [52]. For each category, models included the count of words in the category as the dependent variable, a dummy variable representing the speaker role (participant/facilitator), session number, and group-by-session number interaction as categorical fixed effects. The word count of each contribution was used as an offset variable. To account for heterogeneity in language use across participants and sessions, we included random intercepts for each participant, and each session nested within participants. Adjusted rates of word usage were then calculated for each session and role, along with confidence intervals. 

#### 2.7.3. Agglomerative Hierarchical Clustering on Principal Components (HCPC)

Finally, we undertook agglomerative HCPC to examine patterns of language in the sessions, including all LIWC variables of interest, expressed as a proportion of words belonging to each category per contribution. Variables were standardised prior to analysis.

Preliminary analyses revealed highly skewed distributions for the LIWC variables. For example, the short sentence “Okay, bye” has a 99% Authentic score. To identify such instances, we conducted agglomerative HCPC several times in an iterative process, sequentially identifying and excluding smaller “outlier” clusters including extreme combinations of the variables [53].

To further account for the noisy nature of the data, we conducted principal component analysis (PCA) prior to each clustering iteration and retained the number of dimensions summarising 75% of the total variance, per previous research [54]. We then conducted HCPC on the principal components using the R FactoMineR package (version 2.6) [55], selecting the cluster partition with the higher relative loss of inertia retained as recommended. For each variable, we compared the cluster mean to the overall mean, using a *t*-test accounting for sampling without replacement. Although this method does not allow statistical inference in the context of skewed data, it affords the identification of the most salient or extreme characteristics of each cluster. We considered variables to be specifically descriptive of a cluster if the *p*-value associated with the test was <0.05.

This algorithm allowed a data-driven identification of contribution-level patterns observed in the support group sessions. For granularity, we conducted this process sequentially on the semi-structured and unstructured sessions, and separately for facilitator and participant contributions.

## 3. Results

### 3.1. Descriptive Comparisons to LIWC-Defined Natural Speech Summaries

Summary-variable findings are reported in Figure 1 and highlighted results are presented in Table 5. *p*-values are provided for context and should not be interpreted in terms of statistical significance.

Between-group differences included: the expression of less positive emotion words in the support group corpus (3.25) than in natural speech (5.31), less positive emotional tone in the support group corpus (54.0) than in natural speech (79.29), and more health concerns, especially for participants (support group: 0.95; natural speech: 0.38). Separating out participants and facilitators, participants used fewer first-person singular pronouns (5.19) and second-person pronouns (1.77) than the natural speech corpus averages (7.03 and 4.04, respectively) while using more third-person singular pronouns (2.51 vs. 0.77). Facilitators conversely used more second-person pronouns (4.97 vs. 4.04) and fewer third-person singular pronouns (0.25 vs. 0.77) than the natural speech corpus averages. They also used affiliation words at a higher rate than in the natural speech corpus (3.71 vs. 2.06).

Finally, the initial semi-structured sessions with a pre-set agenda scored higher for analytic language and lower for authentic language than the following unstructured more peer support-focused sessions.

### 3.2. Changes in Language Use across Sessions: Mixed-Effects Poisson Models

Adjusted incidence rates per contribution for each LIWC summary variable are presented in Figure 2.

In all models, including random effects accounting for heterogeneity between individuals and sessions provided a better model fit according to likelihood ratio tests. This suggests that a proportion of the variation in language use observed in the sessions can be attributed to differences between the ways in which group members speak. After adjusting for this heterogeneity, the regressions presented in Figure 1 revealed similar patterns of language use in the semi-structured sessions and visual inspection of the incidence rates per session suggests that these patterns arise independently in Groups 1 and 2.

The models investigate change over time. They show that facilitators’ speech varied greatly between sessions. This being said, facilitators used more analytic language in Sessions 1 and 3 (centring more around the PLWRD) than in Sessions 2 and 4 (centring more around the participant). As for participants, we observed similar patterns in Group 1 and Group 2 for all four variables. A division emerged between Sessions 1 and 3 (characterised by more analytical and confident language) and Sessions 2 and 4 (characterised by more open and positively emotionally charged language). The unstructured sessions showed a rise in authentic language towards the latter half of the sessions, partially in concert with a lower proportion of analytic language over time.

### 3.3. Clustering: Are there Meaningful Patterns of Language?

For facilitators, the clustering algorithm yielded multiple small clusters (average size of 21 observations). This small cluster size could have been due to the high dimensionality of the sample used to conduct analyses. However, we obtained similar results when re-running the algorithm including a smaller number of variables. We concluded that facilitators’ contributions were highly heterogeneous, and that the algorithm could not identify appropriately descriptive/homogenous patterns of language use.

In comparison, we found distinct patterns of language for participants, 14 meaningful clusters were identified amongst 857 observations, an average of 61 observations per cluster. Upon further inspection of the clusters, we identified three categories of contribution: sharing experience (EXP), self-reflection (SELF), and group processes (GROUP). Each category and cluster are characterised by different combinations in the distribution of the clustering variables. Table 6 and Table 7 provide a description of each cluster and an example contribution.

Despite running the clustering algorithms independently, the three main categories of participants’ contributions were found in both semi-structured and unstructured sessions. The most prevalent cluster category was characterised by long conversational turns and a high rate of personal concerns words and third-person singular pronouns, which we interpreted to be the “sharing experience” cluster (EXP). This category represented 72% and 57% of the word count in the semi-structured and unstructured sessions, respectively. A second category of contributions was characterised by a high occurrence of first-person singular pronouns, open and authentic language (authentic), words belying insight and evaluation (insight), and a lack of positive emotional tone. We interpreted this combination of variables as describing authentic self-reflection from participants (SELF). This category of contributions accounted for 12% of the word count in the initial four semi-structured sessions and 37% of the word count in the ensuing unstructured sessions. Finally, “group processes” clusters (GROUP) were characterised by shorter contributions characterising interactions within the group such as “Thanks”, and “Hello”. This category represented 1% and 4% of the word count in the semi-structured and unstructured sessions, respectively. It is important to note that the exact balance of features in the cluster categories differed between semi-structured and unstructured sessions.

The distribution of clusters was then represented across sessions in clock-style visualisations (Figure 3 and Figure 4).

Semi-structured sessions contained a higher proportion of ‘sharing experience’ contributions. This category of contribution represented between 36% and 68% of all contributions in these sessions. Sharing experience contributions also consistently constituted the highest proportion of the total word count of the three cluster categories (71–96% of word count). The sessions were also all marked by a smaller proportion of ‘self-reflection’ contributions in all sessions, in comparison to shared experience, representing between 9% and 35% of contributions and 2–26% of word count. GROUP contributions represented between 17% and 46% of all contributions in the semi-structured sessions and between 0.3% and 5% of word count).

In comparison, unstructured sessions were more variable linguistically. These sessions contained fewer facilitator contributions than the semi-structured sessions in accordance with the increased focus on peer support. Confluently, group process contributions marked a higher proportion of participants’ speech in the unstructured sessions, between 31% and 46% of contributions, and between 2% and 13% of word count. Unstructured sessions also differed from one another in terms of the relative proportion of self-reflective contributions and sharing experience contributions. For example, in Session 6, self-reflective contributions were the most common contribution type, marking 52% of total participant contributions (72% of word count), while in Session 5 sharing experience contributions were the most common contribution type, marking 45% of total participant contributions, (81% of word count). The remaining sessions shared similar proportions of SELF contributions (23–31%) and EXP contributions (24–34%) but varied in the proportion of the total word count for each contribution type (SELF: 14–47%, EXP: 44–78%).

## 4. Discussion

The results above provide a multifaceted linguistic characterisation of peer interactions within multicomponent support groups. Here we consider each of our research questions and present hypotheses as to the mechanisms of the social support observed.

### 4.1. Linguistic Characterisation of the Support Group Space (Research Question 1)

With the first research question, we investigated the differences between conversation in the support group corpus and the natural speech corpus. Means comparisons yielded one important similarity: the distribution of the past, present, and future tenses. This structural similarity between the two corpora implies that conversation in the support groups did not disproportionately centre on the past or anxieties about the future. Instead, participants engaged with one another and the topics for conversation in the way we are used to in general conversation. They related past events to their and their conversational partners’ present and future contexts, rather than simply taking turns to swap histories without engaging with one another.

There were many differences between the support group corpus and the natural speech corpus. Firstly, the support group sessions were, on average, neutral in tone whereas the natural speech corpus is, on average, of very high emotional tone. In previous contexts, authors have suggested that a neutral tone score may suggest a lack of linguistic expression of emotionality [44,56]. However, here, it more likely reflects a balanced range of emotional expression. The support group sessions spent more time engaging in difficult conversations than the natural speech corpus likely did. Accordingly, the positive emotional tone was a grouping characteristic for the group process contributions in the support group corpus, indicating that the other two categories of contribution, sharing experience and self-reflection, were not highly positive in their emotional tone. Positive emotion seemingly was present in response to others sharing rather than representing the default emotional valence for contributions. In most conversational contexts in English, a positive emotional tone is present as the default setting. We suggest that the support groups allowed participants to contribute in ways that contravene the normal conventions of polite speech and focus on negative emotions as well as positive. LIWC has been shown to accurately identify positive and negative emotions in language [46,57]. These findings from previous work support the notion assumed here that LIWC findings do reflect the participant’s emotional state. Taking LIWC as an accurate identifier of emotion, we argue that the importance of and engagement in support groups is underlined by the significantly more negative tone than natural speech.

The balance of personal pronoun focus is another dimension of difference. A primary objective of the groups was to allow care partners of PLWRD to focus on issues surrounding their own identity and independence as well as issues affecting the PLWRD and their relationship to the PLWRD. The gravitational pull of the group’s unifying factor (supporting a PLWRD) was felt in the higher rates of third-person pronouns and first-person plural pronouns than in the natural speech corpus. Participants spent more time referring directly to the PLWRD by their relevant pronouns and indirectly by conjoining themselves into a ‘we’, an ‘our’, or an ‘us’ as the joint focus of the sentence. This is matched by a lower focus by the speaker on themselves alone (first-person singular pronouns) and on the person(s) with whom they are speaking (second-person pronouns). Participants also mentioned health concerns at a higher rate than in the natural speech corpus, further indicative of the goals of the session.

Finally, the speech in the support group corpus is characterised by a higher Analytic score than natural speech, signifying a higher proportion of logical, formal speech [42], a higher Clout score suggesting greater expression of expertise [41], and a lower Authentic score, suggesting a slightly more guarded and distanced style [43]. We offer the interpretation that participants engaged with the topics for discussion, slightly tentatively, but by bringing their wealth of experience and a desire to learn from each other’s knowledge.

In short, it is possible to distinguish the support group space from natural speech, even at the transcript level. The differences in the relative rates of use of personal pronouns and increased focus on health concerns plausibly further distinguish care partner support groups from other types of support groups. We suggest that the lower emotional tone, specifically the relative dearth of positive emotional words, and higher levels of analysis and cognitive processes are a reflection of engagement with the support groups’ topics. This difference from the natural speech corpus highlights how important the opportunity for facilitated peer support can be for underserved populations. To a certain extent, these differences may indicate the space was used properly as a safe place for reflection and sharing for those who normally face many barriers to experiencing targeted support.

The interpretations above are consistent with Social Comparison Theory, with multiple works [58,59,60] highlighting that the experience of uncertainty in several diverse contexts precipitates an increased drive to speak to others in comparable situations and learn from them. The context of rare disease, and of rare dementia specifically, often brings with it feelings of isolation and invitation for downward comparison, including with non-specialised professionals who may have inferior knowledge of the condition. It is both desirable and somewhat unfamiliar for care partners of PLWRD to find themselves among other experts by experience and to have upward comparison and lateral comparison facilitated. As later research questions address, the more time the participants spent in the support groups provided by Rare Dementia Support, the more participants’ language reflected increasing comfort and security in the mutually understanding nature of the groups. The open expression (range of emotional valence) and focus on information sharing and gathering (mention of health concerns, high recounting of issues relating to themselves and PLWRD) further align with Social Comparison Theory’s predictions.

### 4.2. Facilitator and Participant Contributions (Research Question 2)

We posed the second research question to explore how the groups’ facilitators managed the space and how participants explored the topics for discussion and interacted with one another.

The clustering algorithm’s failure to succinctly categorise facilitator contributions in the way it was able to for participants is an important component in understanding their expertise. An interpretation of this is that the facilitators flexibly and expertly adapted their speech to maintain the space for the participants’ consistent use, via insightful reflection of participant contributions and tailored informational support. Facilitators’ offering of outward-looking (fewer first-pronoun pronouns), expert, and crucially more positively emotional contributions to the space aligns with previous research which emphasises the importance of compassion and empathy from facilitators alongside the provision of timely and appropriate informational support and signposting of services [61,62]. Research has suggested that organised maintenance of the support group, balancing the needs of the individual and the group, is important for successful groups [63]. The consistent use of space by participants is evidenced in both the consistent patterns between Group 1 and Group 2 in their approach to the sessions with pre-set agendas and the consistent contribution types identified by the clustering algorithm. 

Participants shifted from being facilitator-led to being increasingly focused on one another as time went on, responding directly to one another more frequently rather than having the facilitator contribute in between each participant. This suggests that participants were increasingly able to both open up about their own difficulties and directly respond to others’. These responses are characterised by kindness, with positive emotion manifesting as an identifying factor in group process contributions. This reflects other findings reporting that the most important ingredients of support groups for care partners were feeling that other care partners were in similar situations and learning from other’s shared experiences [64]. This finding extends to dementia caregivers [65].

Taking both session types together for participants, their contributions are maximally distinguishable by the reference to self and others, the summary variables, and the emotional intensity and valence of the contributions.

Accordingly, we suggest that participants broadly used the group according to its design prerogative. A space was opened to care partners, in which they felt able to share about the recent ongoings in their journeys supporting a PLWRD. Contributions in the sharing experience category frequently went beyond the surface, having a notably higher word count than the other contribution types and covering a breadth of different topic matters (health, money, family). These findings suggest participants were able to open up more fully about how they feel about caregiving for a PLWRD compared with general dementia spaces in which caregiving experiences are often discussed in the context of older age and navigating memory challenges. This is not to say that conversations around health, money, or family will not characterise those spaces too but rather that care partners of PLWRD are less likely to feel they can engage in the conversation as PLWRD have distinct health difficulties, an earlier age of onset [66,67], and different financial and familial challenges. What literature exists underlines the importance of feeling connected to others in the group to valuing the group and returning to it (DLB [68]; FTD [69]; PPA [70]). Studies within the social comparison theoretic research paradigm investigating affiliation have identified uncertainty as a core facilitator [71]. In a wider societal context where care partners of PLWRD often find themselves uncertain of their support options and relative similarity to others, the support groups provided by Rare Dementia Support provided a context to prime affiliation both by satisfying the desire to be amongst others in similar situations [50] and providing the opportunity to talk to other informed care partners who can potentially share information about the shared threat [72].

Further suggestive of the support groups’ utility is the finding that participants felt able to share their own feelings with peers in a highly authentic and reflective manner. Self-reflective contributions scored highly in insightful and evaluative language (insight). The personal narrative has been argued to be a fundamental system for organising perceptions of the self [73] and the sharing of this narrative to sympathetic ears with an understanding of painful dimensions of experience has been suggested to reduce feelings of social isolation and marginalisation [74].

Finally, the identification of sub-categories of group process contributions is notable. The group process contribution clusters cover a range of short functional contributions and a variety of peer-to-peer social engagement. This varied engagement with one another is likely attributable to the shared sensation of experiential similarity, an important factor in the development of mutually beneficial relationships experienced in response to reciprocal emotional and social support [75].

### 4.3. Impact of Session Structure and Theme (Research Question 3) 

We posed the third research question to understand the effect of session agenda and session type on the participants’ contributions and thereby the effect on organisation of thought.

The initial four sessions had a preset agenda to discuss different dimensions of care partner and PLWRD identity and independence. The sessions were designed to precipitate input from peers and facilitators alike on each of these dimensions, focusing on experience-led guidance and informational and emotional support. The design of these support groups reflected evidence that multicomponent groups are more effective than alternatives for dementia care partners [76].

The insights from the Poisson modelling of the summary variables (Authentic, Clout, Analytic, Tone) [15] suggest that agenda was likely influencing organisation of thought for participants. Participants’ contributions were more open and disclosing in sessions designed to focus on themselves than they were in sessions where the focus was placed on the PLWRD (Authentic), where their discourse was more guarded. An interpretation is that, for many, discussing complicated emotions [56,77] towards a loved one without that person present is plausibly complicated [78], especially against the wider societal backdrop of misconceptions of dementia as being synonymous with memory-led symptoms in older people. An alternative explanation is that the care partners wanted to preserve the dignity of the PLWRD and felt it harder to openly discuss their relationship than to discuss issues directly affecting themselves.

The pattern for Authentic Is mirrored in Clout, where a high measure represents confidence and expertise, and a low score reflects humility and tentativeness. Participants spoke with more confidence in sessions focusing on the PLWRD than they did in sessions focusing on themselves. This measure suggests that while participants are experts by experience in the realm of supporting a PLWRD, they spoke less confidently about the issues pertaining to their own identity and independence. When married with the finding presented above that care partners spoke more authentically and openly in sessions designed to focus on themselves, we came to the interpretation of a greater uncertainty in their evaluations of the issues affecting their own identity and independence. It is possible that further specialisation of the support group to specific rare dementias would have precipitated less guarded sharing, but we endorse an alternative explanation that care partners of PLWRD may find opportunities to focus on themselves constricted in the face of caring responsibilities and/or more general emotional pressure [79] and may therefore be more expert in their understanding of issues pertaining to the person they support than themselves.

The findings also suggest that the presence or absence of an agenda affected participants’ language. While there are broad similarities between the clustering categories for the semi-structured and unstructured sessions, there are differences both in their construction, and in the proportion of sessions’ contributions they make up. Group process clusters represented 29% of participants’ contributions in the sessions with an agenda, but 40% of contributions in those without. Reduced facilitation and a lack of a pre-set agenda led participants to fill and monitor the space more themselves.

This change precipitated the emergence of two group process clusters: one characterised by a social, personal focus alongside positive emotional expression [GROUP-a]; the other characterised by positive, formal language without pronominal focus [GROUP-b]. This division captures the difference between the deeper group processes such as support, acknowledgement, and the more functional group processes such as ‘yes’, ‘okay’, and ‘thank you’.

The characteristics of the self-reflective and sharing experience clusters also differed between the two session types. In the semi-structured sessions, higher words per sentence was characteristic of the sharing experience contributions whereas in the continuing group, this variable was higher in the self-reflective contributions. This fits with the greater number of words expended on this topic (37% of unstructured sessions vs. 12% of semi-structured sessions with an agenda) and the different balance of contributions, with self-reflective contributions being more prevalent than sharing experience contributions in half of the sessions without an agenda compared with none of the sessions with an agenda. Words per sentence has previously been found to predict the elaborative processing precipitated by shared attention [80], a finding which raises the possibility that focus was shifted to issues of the self for group members via the joint attention to their own identity and independence facilitated in the group. It also underlines that the longer members spent in each others’ company, the deeper they were willing to look inward, perhaps due to increased confidence in experiential similarity between group members (e.g., [81]).

A further interesting shift is in topic focus. The self-reflective cluster was by far the most negatively emotionally charged of the semi-structured groups’ clusters, however in the unstructured sessions, negative emotion was not one of its primary defining characteristics. The use of insight words has been suggested to reflect reappraisal [17] and has been linked to better health outcomes [47,82]. Insight words are highly associated with the self-reflective cluster in both session types, raising the possibility that expression of thoughts centring around the self became less intrinsically negative but more authentic, more narratively driven, and more focused on the present as participants continued to focus on these issues.

Work testing the disposition for affiliation [72] has highlighted the influence of stress as a facilitator. In stressful contexts, people often turn to affiliation to diminish their stress rather than compare their emotional states with others [83]. We plausibly see this effect here. Care partners of PLWRD often experience high levels of stress and when offered the opportunity for affiliation, they reach for it, increasingly responding to one another with kindness and opening up more and more about themselves. In doing so, they confront the difficulties associated with being a care partner of a PLWRD and redefine the attached emotional valence to their experiences, rather than predominantly swapping externally focused experiences.

### 4.4. Limitations and Future Directions

LIWC2015 has two principal limitations in emotional literacy. The first constraint is the insensitivity to word order [84] making the software unable to read surrounding words to inform the meaning of the target word. For example, the sentence “I am not happy.” receives a high positive emotion score. The second constraint is the few emotion categories, reducing the emotional range the programme can capture. There are no categories for surprise, shame, joy, etc. This being said, these constraints have not prevented LIWC’s effective utilisation in emotion-concerned psychological investigations [46,57]. Alternative programmes with a richer emotional category list and a negation function, such as SEANCE [85], were not purposely designed for psychological inquiry, but rather being trained using movie review corpora. Comparison of such programmes with LIWC in psychological research would constitute an important direction for future comparative research.

Secondly, features that we have interpreted as indicative of a care partner support group might characterise care partner conversation generally. However, this potential criticism does not reflect a limitation for the service provided. Opportunities for care partner to care partner open discourse about personal issues are rare [75]. Furthermore, opportunities afforded to care partners supporting people living with memory-led Alzheimer’s disease are more targeted and suitable than any offered to people caring for PLWRD [79,86], compounding the problem.

Future research should consider the replication of current findings with a larger sample size to test our interpretations. Further comparison of support group corpora to natural speech would be insightful. We expect expert facilitators, whether by experience or by profession, to mirror the current facilitators’ linguistic styles. Participants meanwhile are likely to be introspective and reflect on those they support. The differences are expected to extend to other care partner support group contexts. As here, agenda-based differences between sessions may be uncovered via Poisson modelling of linguistic features. Capturing the linguistic styles of participants in support groups may be of practical use to therapists, counsellors and others facilitating such groups during their training, as they learn and understand how to create safe online spaces for these important conversations.

Online support groups can offer cost-effective opportunities for individuals affected by rare diseases for whom localised support is unlikely to be designed with their exact support needs in mind. By virtue of being online, in-the-moment linguistic analysis is made easier due to the option of automatic transcription. Facial emotional analysis is also made more attainable via video recording. The combination of these two analysis types would be an interesting direction for future work.

Should researchers choose to use similar clustering methods, future research on care partner support groups might find that contributions centre around three central functions: (1) sharing in shared experience, (2) use of shared experience to trigger introspective reflection, and (3) interaction of group members in supportive and group orienting processes. The linguistic features identifying contribution type will likely be led by the distribution of personal pronouns and words per sentence but the exact featural construction of each cluster will be context dependent. LIWC’s category inventory is powerful but not exhaustive. The use of: (a) more precise programs [85], (b) applying established social support frameworks such as the Social Support Behaviour Code (SSBC) [87], and (c) methods such as qualitative content analysis (which allows for the interpretation of meaning while accounting for context in naturalistic text data) could help to further illuminate the PLWRD care partner experience in future work.

We have chosen to evaluate our interpretations against the paradigm of Social Comparison Theory. The findings support the intuition that the support groups offered an opportunity for participants, afflicted by poor experiences of support, to find knowing spaces providing relevant information and experience. When presented with such a space, participants sought comparison and affiliation via sharing with one another and found opportunities for upward comparison in others and in their expert facilitators. This is pertinent in the context of downward comparison being more prominent in populations facing a health threat [88], and in the context of health behaviour as an important realm of expansion for social comparison research [71]. Future work should continue to consider support groups as an important avenue for social comparison and further integrate the theory into methodology. The current findings also contribute to our understanding of how to develop future tools to allow researchers and facilitators to identify, encourage and understand the different types of social support occurring between participants in online support groups. Such a tool could combine the current quantitative linguistic methods with support codes identified by human qualitative researchers, as well as physiological metrics such as facial emotion recognition.

## 5. Conclusions

To conclude, we have advocated for the importance of specialised spaces for those affected by rare disease, and the key role computer-mediated interventions can play in their provision. The unstructured peer-support style group continues to the present day, a strong indication of the efficacy of these groups in combating the social isolation that care partners feel.

Secondly, we have discussed, on multiple levels, the mechanisms by which peer–peer support functioned in the groups. Practically, the sharing of experience, reflecting on one’s own emotional state, and responding to others openly and effectively, in concert with expert facilitation underpin the support. We additionally suggest that social comparison theoretic paradigms are useful in understanding the social behaviour of care partners of PLWRD in support groups. And we would welcome future research employing social comparison theory in other support group contexts.

Future work using the presented method in concert with either qualitative analyses or standardised outcome measures can further inform on the potential utility of this method to provide information sitting in the interspace between qualitative and quantitative information. The methodology detailed above, and the development of accompanying tools to facilitate analysis for non-academic audiences, may allow service providers to glean meaningful insight into the use of their groups in the absence of or complementary to standardised outcome measures.

## Figures and Tables

**Figure 1 healthcare-12-00313-f001:**
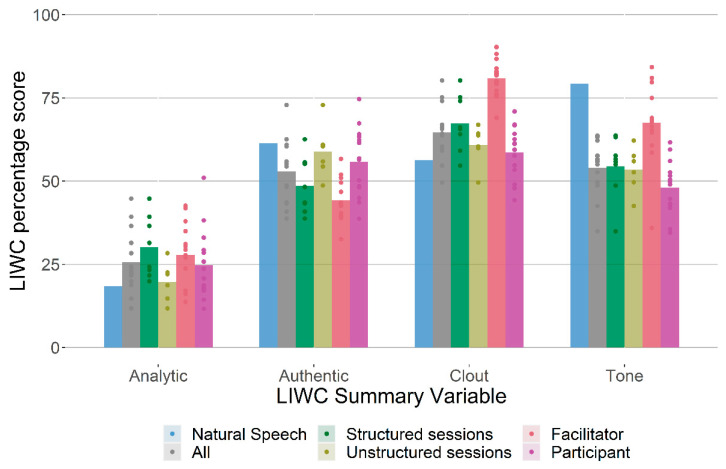
LIWC Summary variables, mean transcript score for present corpus and for the LIWC Natural Speech corpus. Note: Each circle represents a session transcript (*n* = 14; 8 semi-structured, 6 unstructured).

**Figure 2 healthcare-12-00313-f002:**
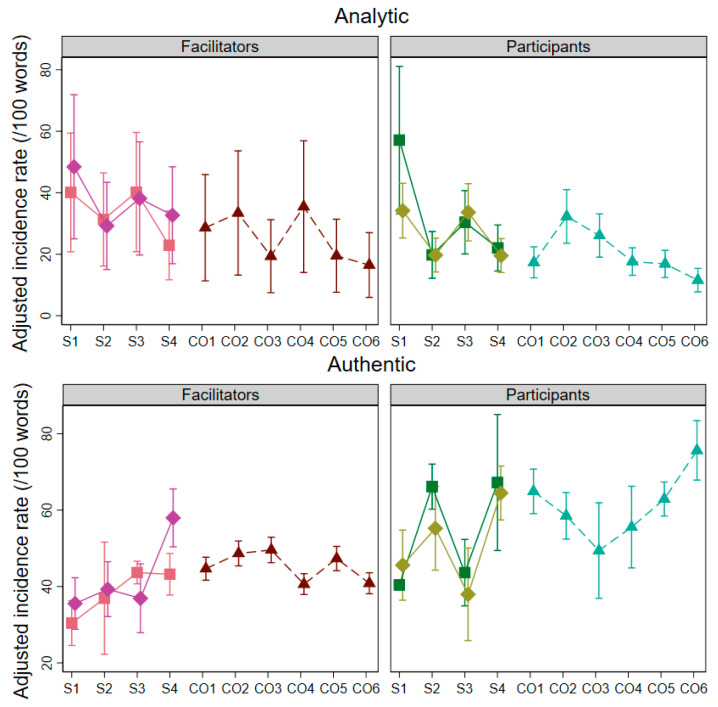
Adjusted incidence rates per contribution for each LIWC summary variable.

**Figure 3 healthcare-12-00313-f003:**
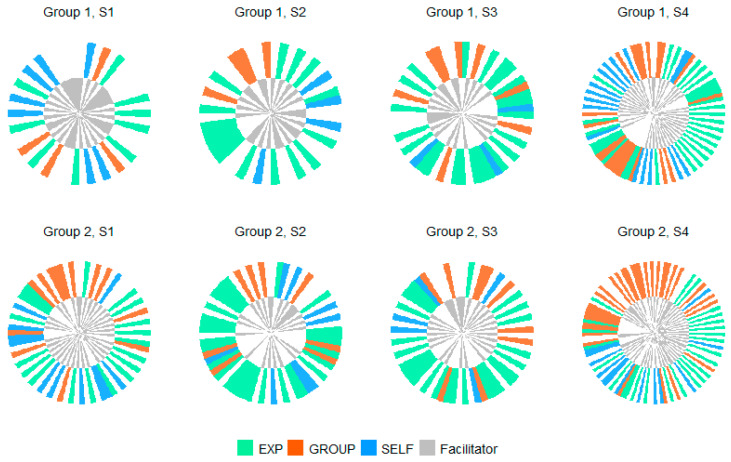
Clock-style visualisation of cluster distributions between sharing experience (EXP), self-reflection (SELF) and group processes (GROUP) in semi-structured sessions. Note: 1 bar = 1 conversational turn.

**Figure 4 healthcare-12-00313-f004:**
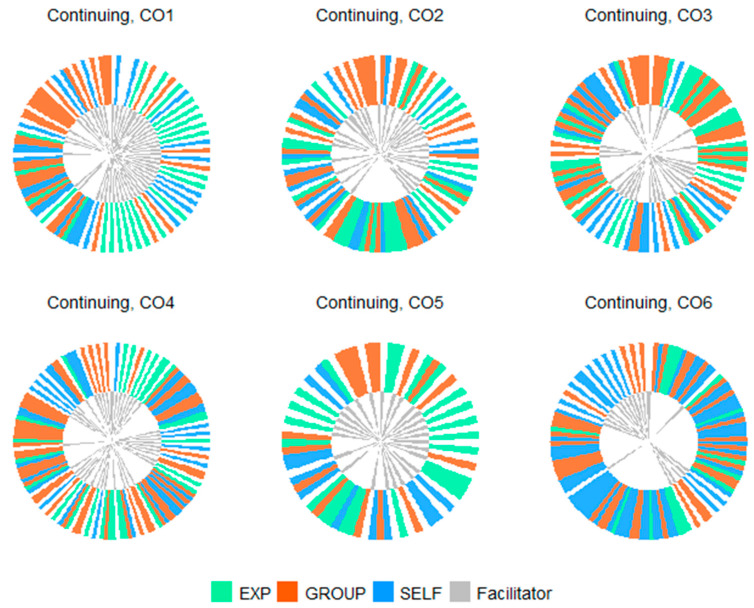
Clock-style visualisation of cluster distributions between sharing experience (EXP), self-reflection (SELF) and group processes (GROUP) in unstructured sessions. Note: 1 bar = 1 conversational turn.

**Table 1 healthcare-12-00313-t001:** Text corpus, data available.

Transcript ID/Session ID	Theme	Attendees	N Contributions; N Words
Semi-structured Sessions (Group 1)			
#1/G1S1	S1—Personal introductions	2F, 5P	N_contributions_ = 54; N_words_ = 10,259
#2/G1S2	S2—Care partner independence	2F, 2P	N_contributions_ = 48; N_words_ = 9282
#3/G1S3	S3—PLWRD identity and independence	2F, 3P	N_contributions_ = 57; N_words_ = 9247
#4/G1S4	S4—Care partner identity and group conclusion	2F, 3P	N_contributions_ = 109; N_words_ = 10,143
Semi-structured Sessions (Group 2)			
#5/G2S1	S1—Personal introductions	2F, 6P	N_contributions_ = 77; N_words_ = 11,204
#6/G2S2	S2—Care partner independence	2F, 5P	N_contributions_ = 68; N_words_ = 8805
#7/G2S3	S3—PLWRD identity and independence	2F, 5P	N_contributions_ = 68; N_words_ = 8521
#8/G2S4	S4—Care partner identity and group conclusion	2F, 5P	N_contributions_ = 127; N_words_ = 8631
Unstructured/Continuing Sessions			
#9/CO1	Participant directed discussions	1F, 5P	N_contributions_ = 132; N_words_ = 8415
#10/CO2		1F, 7P	N_contributions_ = 111; N_words_ = 8868
#11/CO3		1F, 5P	N_contributions_ = 118; N_words_ = 8828
#12/CO4		1F, 6P	N_contributions_ = 150; N_words_ = 9507
#13/CO5		1F, 6P	N_contributions_ = 77; N_words_ = 8889
#14/CO6		1F, 3P	N_contributions_ = 124; N_words_ = 9617

Note: Abbreviations: F = Facilitator, P = Participant.

**Table 2 healthcare-12-00313-t002:** Included LIWC Summary variables.

Variable (Abbreviation)	Description (Examples)
Summary Variables	
Clout	High score: high expertise, confident/Low score: tentative and humble [41]
Analytic	High score: formal, logical, hierarchical/Low score: informal, personal, here-and-now [42]
Authentic	High score: honest, personal, disclosing/Low score: guarded, distanced [43]
Tone	High score: positive, upbeat/Low score: anxious, sad, hostile [44]

**Table 3 healthcare-12-00313-t003:** Included LIWC linguistic variables.

Variable (Abbreviation)	Description (Examples)	
Linguistic Dimensions		
*Personal Pronouns—inform on the locus of attention and the centre from which it projects* [45]		
1st person singular (I)	1st person singular pronouns (I, me, mine)	
2nd person (You)	2nd person pronouns (you, your, yours)	
3rd person singular (Shehe)	3rd person singular pronouns (she, him, herself)	
1st person plural (We)	1st person plural pronouns (we, ourselves, us)	
3rd person plural (They)	3rd person plural pronouns (they, them, their)	
*Tense Focus—indicates the time frame to which the speaker is directing attention* [17]		
Present focus (focuspast)	Words organising attention to:	the present (today, is, now)
Past focus (focuspresent)		the past (ago, did, talked)
Future focus (focusfuture)		the future (may, will, soon)

**Table 4 healthcare-12-00313-t004:** Included LIWC psychological variables.

Variable (Abbreviation)	Description (Examples)	
Psychological Dimensions		
*Affective Processes—reveal emotional content and the affective stance an individual is taking* [46]		
Affective processes (affect)	Words carrying emotional charge (happy, cried)	
Positive emotion (posemo)	Positive emotionally charged words (love, nice, sweet)	
Negative emotion (negemo)	Negative emotionally charged words (hurt, ugly, nasty)	
Anxiety (anx)	(worried, fearful)	
Anger	(hate, kill, annoyed)	
Sadness (sad)	(crying, grief, sad)	
*Cognitive Processes—belie the way individuals organise their thoughts and the strength with which they hold their beliefs* [47]		
Cognitive processes (cogproc)	Words conveying consideration and manipulation of ideas (cause, ought, know)	
Insight	Words conveying consideration of ideas (think, know)	
Causation (cause)	(because, effect)	
Tentative (tentat)	(maybe, perhaps)	
*Personal concerns and motivations—inform on the individual’s motivations, impressions of others, and further develop the external foci of the individual* [48]		
Drives	Words pertaining to:	different motivations (bad, kinship, try)
Affiliation		positive social processes (ally, friend, social)
Work		work (job, majors, xerox)
Home		the home (kitchen, landlord)
Health		health (clinic, flu, pill)

**Table 5 healthcare-12-00313-t005:** Summary of similarities and differences between support group sessions and LIWC’s natural speech corpus, by type of session (semi-structured or unstructured) and role of contributor (participant or facilitator).

Summary of Findings		LIWC Variable	I&I % Mean	NS % Mean	*p*-Value I&I vs. NS	Cohen’s d
	*Similarities between sessions and the natural speech corpus*					
All vs. natural speech	Similar use of tenses	Focuspast	4.28	3.78	0.081	0.505
		Focuspresent	14.18	15.28	0.085	−0.498
		Focusfuture	1.78	1.45	0.023	0.688
	*Differences between sessions and the natural speech corpus*					
All vs. natural speech	Less positive emotional Tone	Tone	54.02	79.29	<0.001	−3.163
	More first-person plural pronouns	We	1.64	0.87	<0.001	1.328
	Less first-person singular pronouns	I	4.48	7.03	<0.001	−2.318
	More Cognitive processes words	Cogproc	15.30	12.27	<0.001	2.780
	Less affective process words (due to less positive emotions words)	Affect	4.70	6.54	<0.001	−4.982
		Posemo	3.25	5.31	<0.001	−4.905
	*Differences specific to participants and facilitators, compared to the natural speech corpus*					
Participants vs. natural speech	Less second singular person pronouns	You	1.77	4.04	<0.001	−3.603
	More third-person singular pronoun	Shehe	2.51	0.77	<0.001	1.740
	More health and less work personal concerns	Health	0.95	0.38	<0.001	1.295
		Work	1.38	2.87	<0.001	−3.921
Facilitators vs. natural speech	More Clout language	Clout	80.93	56.27	<0.001	4.419
	Less third-person singular pronouns	Shehe	0.25	0.77	<0.001	−2.080
	More use of second-person pronouns	You	4.97	4.04	<0.001	1.788
	More affiliation words	Affiliation	3.71	2.06	<0.001	1.435
	Less personal concern words overall	Work	1.28	2.87	<0.001	−4.077
		Home	0.14	0.34	<0.001	−2.222
	*Differences specific to structured and unstructured sessions, compared to the natural speech corpus*					
Structured sessions vs. natural speech	More Analytic language	Analytic	32.88	18.43	<0.001	1.268
	Less Authentic language	Authentic	41.82	61.32	<0.001	−1.520
Unstructured sessions vs. natural speech	No *additional* notable differences.					

**Table 6 healthcare-12-00313-t006:** Cluster composition and representative contributions, semi-structured sessions.

Cluster Name (N) Cluster Description	Most Representative Contribution(s)
*EXP—SHARED EXPERIENCE CLUSTERS (N = 177 contributions, 87% total word count)*	
EXP-a (N = 134) Elaborations on personal concerns	“Our walks together have been limited by what [NAME] can do. I mean, I do not think we realise because we can see everything, when we were going down some steps she said ‘It is although I am spreading out into space. I am stepping into space!’, and she was terrified...”
LIWC descriptors	+++ WPS, health, leisure, money, they, female, family
	--- posemo, you, Analytic
EXP-b (N = 43) Sharing about the PLWRD	“Right, my name is [NAME]. I live in [LOCATION]. My wife is [NAME] and she was diagnosed with PCA […]. She was really the person who flagged it up herself. …” “He has trouble judging time. He knows he has got a watch that tells the time, but he often gets the alarm. And we have had chucks going off at night-time because it has got muddled up.”
LIWC Descriptors	+++ shehe, male, female, family, WPS
	--- Authentic, you, Tone, focuspast, affect, posemo
*SELF—SELF-REFLECTIVE CLUSTERS (N = 67 contributions, 12% total word count)*	
SELF (N = 67) Insightful authentic self-reflection	“[…] I try as much as possible to keep in the day, because I honestly don’t know what tomorrow is going to bring. It could be a good day. It could be a bad day. I think the worst part about it is to project forward. During the peak of COVID-19 I was absolutely terrified.”
LIWC Descriptors	+++ I, Authentic, insight, sad
	--- Clout, social, Tone, you, posemo, shehe, we
*GROUP—GROUP PROCESSES CLUSTERS (N = 102 contributions, 1% total word count)*	
GROUP-a (N = 64) Short group-building social contributions	“Where are you based, [NAME]” “Okay, I appreciate it. Yes. Nice to meet you all. Ok, thanks a lot, [NAME].” “Both of you. No both of you, you’ve done a lot.”
LIWC Descriptors	+++ Analytic, social, work, early contributions, work
	--- WPS, cogproc, tentat, focuspast, we
GROUP-b (N = 26) Short, positive statements	“Take care. Thank you. Bye.” “Well done.” “Thank goodness you’re doing that research.”
LIWC Descriptors	+++ posemo, you, focuspresent, Tone, social
	--- Authentic, WPS, tentat, I, drives
GROUP-c (N = 1) Short negative statement	“Sorry”
LIWC Descriptors	+++ sad, negemo
GROUP-d (N = 8) short group-building contributions	“Hi, [NAME].” “Very helpful.”
LIWC Descriptors	+++ affiliation, drives, we, social, risk, Clout
	--- Authentic, WPS, cogproc

Note: +++ average scores for variables cited are higher in cluster compared to all contributions, --- average scores for variables cited are lower in cluster compared to all contributions.

**Table 7 healthcare-12-00313-t007:** Cluster composition and representative contributions, unstructured sessions.

Cluster Name (N) Cluster Description	Most Representative Contribution(s)
*EXP—SHARED EXPERIENCE CLUSTERS (N = 141 contributions, 57% total word count)*	
EXP-a (N = 140) Lengthy sharing concerning the PLWRD, especially on the topics of health and family	“Dementia is bad enough with the confusion anyway without sending a new set of people, and also social services who fund it. […] he doesn’t want four different people coming in for half an hour every day. It’ll really confuse him […]”
LIWC Descriptors	+++ shehe, health, male, family, focuspast, female, friend
	--- you, less early contributions
EXP-b (N = 1) Short comment on PLWRD	“She’ll cope.”
LIWC Descriptors	+++ female, shehe, focusfuture, social, Clout
	--- focuspresent, Authentic, cogproc Analytic
*SELF—SELF-REFLECTIVE CLUSTERS (N = 166 contributions, 37% total word count)*	
SELF (N = 166) Insightful, authentic self-reflection	“I knew I was grieving for my life because I could say that, but I still didn’t think about it as a grief. Or probably you get, sometimes it pulls you back down and then you get up again and go on…” “I’m a walking disaster, an almost not-walking disaster. So, it’s true, you’ve got to look after yourself…”
LIWC Descriptors	+++ I, Authentic, death, insight, anx
	--- Clout, Analytic, social, you, Tone, posemo
*GROUP—GROUP PROCESSES CLUSTERS (N = 102 contributions, 6% total word count)*	
GROUP-a (N = 109) Group-building social processes	“Yes, hopefully it helps.” “Yes, you’re very positive.”
LIWC Descriptors	+++ social, you, Clout, they, focuspresent, money, focusfuture, posemo, drives
	--- WPS
GROUP-b (N = 91) Short positive social contributions	“Yes, interesting.” “oh, happy birthday!” “Absolutely.”
LIWC Descriptors	+++ Analytic, posemo
	--- WPS, social, Clout, Authentic, I, you, tentat, drives
GROUP-c (N = 2) Short ‘negative’ statements	“Sorry.”
LIWC Descriptors	+++ sad, negemo, Analytic
GROUP-d (N = 5) Short negative reflections	“Men cry too.” “Oh, that’s a shame.”
LIWC Descriptors	+++ male, negemo, anxiety, risk, sad, shehe
	--- Tone, Authentic, WPS, focuspresent

Note: +++ average scores for variables cited are higher in cluster compared to all contributions, --- average scores for variables cited are lower in cluster compared to all contributions.

## Data Availability

Data and code will be made available upon conclusion of the grant under which this project has been undertaken (ES/S010467/1) in line with the stipulations in the study protocol (Brotherhood et al., 2020) [35]. At the completion of the grant the pseudonymised data relevant to this publication will be uploaded to a data repository and the code will be made available on GitHub.

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
