# Peer review of "How Do Care Partners of People with Rare Dementia Use Language in Online Peer Support Groups? A Quantitative Text Analysis Study"

_healthcare, 2024, doi:10.3390/healthcare12030313_

Round 1
Reviewer 1 Report
Comments and Suggestions for Authors
Analysis of social support and upward and downward comparison in support groups is an important and interesting topic. The use of AI and text analysis is exciting. However, this paper is very difficult to read. The combination of V E R Y long sentences, passive voice, use of academic language specific to this field by experts is language use made me wonder what audience you were targeting the paper to. Did you want it to be helpful to the rare disease community, caregivers, people who run support groups or just those interested in quantitative analysis? The writing really reminded me of a 2020 article “Science is getting harder to read” https://www.nature.com/nature-index/news/science-research-papers-getting-harder-to-read-acronyms-jargon#:~:text=Long%20titles%2C%20longer%20abstracts&text=Not%20only%20do%20overly%20wordy,were%20just%2010%20words%20long. I also wondered if the facilitators of the group read this paper and whether they agreed what you deduced from text analysis.
The Abstract is a good summary of the paper. Despite the long and somewhat convoluted sentences, the introduction clearly makes a case for this research. The citations are relevant to the background for this study. The study design is clearly presented and can be used by understand and replicate what was done.
Findings and Discussion
Your finding resonate with my experience as someone very familiar with caregiving, caregiving research, caregiving support groups, and social support, and that upward and downward comparison is often the reason that caregivers drop out of support groups, e.g., “They were all talking about how bad it is to have a husband with dementia…..” or “It was just a bitch session”
However, I don’t think that as a reader that I should have to keep referring back to find out what all the acronyms mean or read long sentences a couple of times to try and figure out what you mean. I really wondered who was “it” in the, “It was concluded,” line 311. Do you really want to use passive voice when active voice is much more readable? Be careful with use of words that carry a particular value eg., boasted in line 354 in your findings. Also, some people being cared for are not “loved ones” or even liked ones. There are some divorced spouses who will care for a person they divorced because they feel obligated. I suggest that whatever term you decide to use person living with a rare disease, care recipient, PLWRD, use it consistently throughout for clarity.
This is very worthwhile research that I think is obscured in the writing. I will be interested to see if the other reviewers concur. I am attaching a PDF of my notes as a read through it. I had to print it just so I could refer back and forth to the acronyms.
Additional comments:
I think readability depends on who they are targeting this paper to. It is absolutely fine if they are only targeting the audience using quantitative text analysis. If they are looking for a broader audience-- particularly those interested in support groups and types of support offered in groups it needs to be revised to be more readable.
Many of the sentences are run-on. e.g., "The study’s data source was a 14-transcript text corpus of small group discussions centring around independence and identity, set up by Organisation A in response to growing support demands (10, 11, 37) in the face of the COVID-19 pandemic, recorded between July 2020 and February 2021.
Clearer language--Use one term and define it. A good example is caregivers, they are familial caregivers line 14, informal caregivers line 30 Similarly with people living with rare dementia Identify a clear actor. Who is doing what? and avoid nominalizations perhaps multi component support groups by video conference line 66 67- In the present study, multi-componential video-conferenced support groups set up in the service of the aforementioned population in response to the pandemic are examined. Keep subjects and verbs close together and revise for conciseness. Line 42. Ever expanding technological capabilities and access coupled with a shared global health crisis brings the topic into focus as of central importance for serving public health. could be rewritten to The topic is brought into public health focus by expanding technological capability and access to technology coupled with a global health crisis. Can this be written to improve the clarity? Line 52 to 54 "Whilst there is debate as to the relative utility of online support groups compared with face-to-face interventions, there is a need for online support and studies have shown both forum-style opportunities (e.g., 7) and videoconferencing groups (e.g., 8) to exert positive effects." Line 56 is "form" the best verb to use in this sentence? Caregivers of individuals affected by rare forms of dementia (PLWRD) form no exception. Caregivers are no exception. Line 59 Can you revise for conciseness? Remove unnecessary words such as only and reduce the number of ands? Perhaps you can relate the isolation to support restriction. The COVID-19 pandemic only heightened the sensation of isolation in caregivers and PLWRD and emphasised geographical isolation via the implementation of lockdowns and increased hesitancy for travel.Whilst only exacerbating the support needs of those affected by a rare dementia, the pandemic greatly restricted previously available support. I do not have time to go through this entire paper and make suggestions for readability. If this was my paper, I would make an appointment with the University writing service. I am sure are guides at this group's university, this is an example from the U of Calgary https://www.ucalgary.ca/live-uc-ucalgary-site/sites/default/files/teams/9/10-strategies-for-readable-writing-writing-strategies.pdf
This paper could have a wide readership but the very academic, highly technical style make it difficult to read.
Author Response
1. Summary
Thank you very much for taking the time to review this manuscript. Please find the detailed responses below and the corresponding revisions/corrections in track changes in the re-submitted files. We found all of the suggested changes about readability very helpful and in addressing these we have collectively made over 2,500 changes to the text, all of which can be seen in tracked changes.
2. Response to Comments on the Quality of English Language
Point 1: English very difficult to understand/incomprehensible
Response 1: Many of the 2500 total revisions have aimed to make the English more easily understandable to broaden the audience of the paper and lessen the cognitive load on the reader. We hope the reviewer finds our edits acceptable and that the paper is now more easily comprehensible.
3. Point-by-point response to Comments and Suggestions for Authors
Comments 1: this paper is very difficult to read. The combination of V E R Y long sentences, passive voice, use of academic language specific to this field by experts is language use made me wonder what audience you were targeting the paper to. Did you want it to be helpful to the rare disease community, caregivers, people who run support groups or just those interested in quantitative analysis?
Response 1: We thank the reviewer for highlighting that the paper could reach a wide audience if the language were simplified. We have endeavoured to simplify the sentence structure and terminology throughout the paper in an effort to better engage with prospective readers from the rare disease community, care partners, and individuals who run support groups.
Comments 2: I also wondered if the facilitators of the group read this paper and whether they agreed what you deduced from text analysis.
Response 2: Thank you for highlighting that we hadn’t made this clear. SC and EH were the facilitators for the groups and are coauthors on the current paper, and this has now been explicitly mentioned on line 203. They both provided their support for the interpretations we present.
Comments 3: Despite the long and somewhat convoluted sentences, the introduction clearly makes a case for this research.
Response 3: We thank the reviewer for their appraisal of the evidence we provide in support of the research. In addressing comment (1.) we hope to have removed and replaced convoluted sentences.
Comments 4: However, I don’t think that as a reader that I should have to keep referring back to find out what all the acronyms mean or read long sentences a couple of times to try and figure out what you mean.
Response 4: We agree with the reviewer that needing to reread sentences to understand the intended meaning is obstructive to the manuscript’s readability and have taken steps throughout the entire paper to mitigate this. We have also replaced many acronyms with their full versions to ease readability such as the cluster names.
Comments 5: who was “it” in the, “It was concluded,” line 311. Do you really want to use passive voice when active voice is much more readable?
Response 5: Thank you for raising this specific detail. We have updated the use of the passive voice to the active voice to aid readability.
Comments 6: Be careful with use of words that carry a particular value eg., boasted in line 354 in your findings.
Response 6: Thank you again for your attention to detail, we have changed the wording here and considered the choice of words throughout carefully.
Comments 7: Also, some people being cared for are not “loved ones” or even liked ones. There are some divorced spouses who will care for a person they divorced because they feel obligated. I suggest that whatever term you decide to use person living with a rare disease, care recipient, PLWRD, use it consistently throughout for clarity.
Response 7: We are especially grateful for this comment as it reminds us of the breadth of the caregiving population and of the importance in careful consideration of terminology in all sections of the paper. This example has been replaced and all terminology has been checked for consistent use.
Comments 8: I think readability depends on who they are targeting this paper to. It is absolutely fine if they are only targeting the audience using quantitative text analysis. If they are looking for a broader audience-- particularly those interested in support groups and types of support offered in groups it needs to be revised to be more readable.
Response 8: We have no intentions of confining the potential readership of this paper to a specialist audience. We have made substantial changes to the level of description in multiple places to widen accessibility and increase the impact of this manuscript’s findings, which are applicable to practitioners as well as academic researchers.
Comments 9: Many of the sentences are run-on. e.g., "The study’s data source was a 14-transcript text corpus of small group discussions centring around independence and identity, set up by Organisation A in response to growing support demands (10, 11, 37) in the face of the COVID-19 pandemic, recorded between July 2020 and February 2021.
Response 9: Again, we thank the reviewer for their specific and actionable comments about readability, we have made efforts to remedy this dimension also. The example given now reads as: "The study’s data source was a text corpus of transcripts of 14 support group discussions centring around independence and identity [36]. Organisation B set these support groups up as part of their response to growing support demands [10,11,37] caused by the COVID-19 pandemic. The support group sessions in the current corpus were recorded between July 2020 and February 2021."
Comments 10: Clearer language -- Use one term and define it. A good example is caregivers, they are familial caregivers line 14, informal caregivers line 30 Similarly with people living with rare dementia Identify a clear actor. Who is doing what?
Response 10: As mentioned in (7.) above, we have made great efforts to make all of our language clearer and consistent and have defined terminology at a much higher rate, we have exclusively referred to the participants as participants or care partners of a PLWRD (person living with a rare dementia) for example. Should we have missed anything, we would be grateful for the chance to remedy again.
Comments 11: Avoid nominalizations perhaps multi component support groups by video conference line 66 67- In the present study, multi-componential video-conferenced support groups set up in the service of the aforementioned population in response to the pandemic are examined.
Response 11: We have amended the passage to the structure proposed by the reviewer.
Comments 12: Keep subjects and verbs close together and revise for conciseness. Line 42. Ever expanding technological capabilities and access coupled with a shared global health crisis brings the topic into focus as of central importance for serving public health. could be rewritten to The topic is brought into public health focus by expanding technological capability and access to technology coupled with a global health crisis.
Response 12: We have amended the passage containing the example and hope that the new structure fits the reviewers’ considerations. We have also amended our sentence structure throughout to be simpler.
Comments 13: Can this be written to improve the clarity? Line 52 to 54 "Whilst there is debate as to the relative utility of online support groups compared with face-to-face interventions, there is a need for online support and studies have shown both forum-style opportunities (e.g., 7) and videoconferencing groups (e.g., 8) to exert positive effects."
Response 13: We have amended the sentence to improve readability and hope the reviewer finds the new formulation more appropriate.
Comments 14: Line 56 is "form" the best verb to use in this sentence? Caregivers of individuals affected by rare forms of dementia (PLWRD) form no exception. Caregivers are no exception.
Response 14: We have amended the sentence to improve readability and hope the reviewer finds the new formulation more appropriate. "Another population where this is relevant is people living with rare dementias (PLWRD) – by which we mean atypical (e.g., non-memory-led), young-onset (symptoms/diagnosis before age 65) and directly inherited forms of dementia. Care partners of PLWRD experience the access difficulties outlined above."
Comments 15: Line 59 Can you revise for conciseness? Remove unnecessary words such as only and reduce the number of ands? Perhaps you can relate the isolation to support restriction. The COVID-19 pandemic only heightened the sensation of isolation in caregivers and PLWRD and emphasised geographical isolation via the implementation of lockdowns and increased hesitancy for travel. Whilst only exacerbating the support needs of those affected by a rare dementia, the pandemic greatly restricted previously available support.
Response 15: We have amended the sentence to improve readability and hope the reviewer finds the new formulation more appropriate. "The COVID-19 pandemic heightened the often reported sensation of isolation in PLWRD and their care partners [9] and emphasised geographical isolation via the implementation of lockdowns and increased hesitancy for travel. Whilst exacerbating the support needs of PLWRD and their care partners [10], the pandemic greatly restricted previously available support. Existing in-person support either ceased [11] or was adapted for online implementation, making the effective implementation of online support paramount."
Comments 16: This paper could have a wide readership but the very academic, highly technical style make it difficult to read.
Response 16: We hope to have undertaken the necessary steps to simplify the style in the introductory and discussion sections to widen the readership whilst maintaining the precision and depth of focus of the methods and results sections to ensure the paper is still notable for methodologically focused researchers.
Reviewer 2 Report
Comments and Suggestions for Authors
Thank you very much for the opportunity to evaluate your work, here are some considerations that you should take into account to make your study more attractive to the reader.
To begin with, I find the title very cumbersome and confusing and it does not respond to the main objective of your study, please modify it to respond to the more concrete purpose of the study.
The abstracts do not specify the objective of the study.
The introduction is very long, you talk at some point about "rare" diseases without specifying or making references to these diseases(49) and I think it is important for the reader to know which diseases you are referring to. It would be interesting to remove and summarise those issues that are most important and, if they are the subject of your study, to review them.
At the beginning of the methodology you talk about the approval of the ethics committee but it appears in XXXX... could you explain it?
The selection of a sample in a study is a crucial process that influences the validity and applicability of the findings. Here are some guidelines and considerations you can take into account when choosing a sample to methodologically better specify sample selection, criteria, etc.
example.
Purpose of the study:
Clearly define the purpose of your qualitative study - what specific questions or issues are you trying to explore? Sample selection should be aligned with the objectives of the study.
Purposive sampling:
In qualitative studies, it is common to use purposive sampling, where participants are deliberately selected because of their experience, knowledge or relevance to the phenomenon you are investigating.
Diversity:
Look for diversity in the sample. This may include diversity in terms of age, gender, ethnicity, socio-economic status, education or other characteristics relevant to your research. Diversity in the sample increases the richness of perspectives.
Saturation:
Consider the principle of saturation. In qualitative studies, you want to achieve saturation, which means that you have collected enough data to fully understand the phenomenon in question. Saturation will guide the determination of sample size.
Accessibility:
Assesses the accessibility of participants. Consider the availability and accessibility of individuals who could contribute significantly to your research. Make sure you have access to the population you wish to study.
Inclusion/exclusion criteria:
Establish clear criteria for inclusion and exclusion of participants in your sample. This will help ensure that the participants selected are directly related to the issues you are presenting us with in your research.
Context:
Take into account the cultural and social context. Understanding context is essential in research. Make sure your sample reflects the cultural and social diversity relevant to your study.
Sample size:
Sample size is not determined by statistical formulae, but is guided by the principle of saturation. However, take into account the feasibility of the study and the depth you wish to achieve in your analyses.
Ethics:
Be sure to address ethical considerations and participant selection well.
How did you obtain informed consent, how online will you protect the privacy and confidentiality of participants, and how have these been conducted.
Remember that sample choice is a reflexive and strategic process that is tailored to the exploratory and holistic nature of this type of research.
Another question:
Figures (3-4) and images some are difficult to visualise and do not present the data, you could revise.
The definitions of rare diseases, as well as those of rare disease carers, to whom they refer, should be improved for better understanding?
The feeling is that they have a lot of information, but in some cases they do not go in depth and sometimes saturating the reader with so much information is better to prioritise and give quality to the data.
As you comment at the end of your article, the mechanisms by which peer support worked in the groups have been debated at multiple levels. ? In practice, sharing experiences, reflecting on one's own emotional state and responding to others in an open and effective way, in combination with expert facilitation. Theoretically, social comparison theoretical paradigms as useful for understanding the impulses to be in spaces and to respond positively to those spaces and to seek affiliation within them. could better specify these lines of future or improvement.
it is a very interesting paper, which needs revision on some of the issues i bring to you, and which will surely help the scientific community to understand its usefulness. for this i must congratulate you.
Author Response
1. Summary
Thank you very much for taking the time to review this manuscript. Please find the detailed responses below and the corresponding revisions/corrections in track changes in the re-submitted files. We found all of the suggested changes about readability very helpful and in addressing these we have collectively made over 2,500 changes to the text, all of which can be seen in tracked changes.
2. Response to Comments on the Quality of English Language
Point 1: I am not qualified to assess the quality of English in this paper
Response 1: We are sorry to hear that the English in our paper was too complex and inaccessible to be assessed by Reviewer 2. This is of course not our intention and we have taken great lengths, over 2500 total revisions, to simplify the language and make the paper much more readable.
3. Point-by-point response to Comments and Suggestions for Authors
Comments 1: To begin with, I find the title very cumbersome and confusing and it does not respond to the main objective of your study, please modify it to respond to the more concrete purpose of the study.
Response 1: Thank you for your feedback. We have amended the title to reflect the objective of the work within the manuscript. The new version is: How do care partners of people with rare dementia use language in online peer support groups? A Quantitative Text Analysis Study
Comments 2: The abstracts do not specify the objective of the study
Response 2: This comment was more difficult to reconcile with reviewer 1’s perception that the abstract “is a good summary of the paper”. We have made edits to the abstract to improve readability and hope that these sufficiently address this comment. The new version of the abstract is as follows:
We used quantitative text analysis to examine conversations in a series of online support groups attended by care partners of people living with rare dementias (PLWRD). We used transcripts of 14 sessions (>100,000 words) to explore patterns of communication in trained facilitators’ (n=2) and participants’ (n=11) speech and to investigate the impact of session agenda on language use. We investigated the features of their communication via Poisson regression and a clustering algorithm. We also compared their speech with a natural speech corpus. We found that differences to natural speech emerged, notably in emotional tone (d=-3.2, p<0.001) and cognitive processes (d=2.8, p<0.001). We observed further differences between facilitators and participants, and between sessions based on agenda. The clustering algorithm categorised participants’ contributions into three groups: sharing experience, self-reflection, and group processes. We discuss the findings in the context of Social Comparison Theory. We argue dedicated online spaces have a positive impact on care partners in combatting isolation and stress via affiliation with peers. We then discuss the linguistic mechanisms by which social support was experienced in the group. The present paper has implications for any services seeking insight into how peer support is designed, delivered, and experienced by participants.
Comments 3: The introduction is very long, you talk at some point about "rare" diseases without specifying or making references to these diseases and I think it is important for the reader to know which diseases you are referring to. It would be interesting to remove and summarise those issues that are most important and, if they are the subject of your study, to review them.
Response 3: We hope the reviewer is satisfied with changes made in the introduction. We have shortened the introduction and been more explicit in our mention of rare diseases before narrowing our focus to rare forms of dementia. We aim to emphasise the shared components of the rare dementia experience with other rare diseases. The relevant section of the introduction now reads as:
"The relative utility of online support groups compared with face-to-face interventions is disputed, but studies have shown both forum-style opportunities (e.g., [4]) and videoconference groups (e.g., [5]) to have positive effects. Findings extend to populations affected by rare diseases such as Cystic Fibrosis [6] and Duchenne’s Muscular Dystrophy [7]. The potential advantages of online intervention are amplified for those affected by rare diseases (conditions affecting fewer than 1 in 2,000 people [8]), where targeted support is seldom locally available.
Another population where this is relevant is people living with rare dementias (PLWRD) – by which we mean atypical (e.g., non-memory-led), young-onset (symptoms/diagnosis before age 65) and directly inherited forms of dementia. Care partners of PLWRD experience the access difficulties outlined above."
Comments 4: At the beginning of the methodology you talk about the approval of the ethics committee but it appears in XXXX... could you explain it?
Response 4: We apologize for any lack of clarity on our part in our attempts to thoroughly blind the manuscript for review. The ethics number and awarding body will be reintroduced into the manuscript upon completion of the review process. Other blinded elements are our mention of ‘University A’ and ‘Organisation B’ which will also be unblinded upon completion of the review process.
Comments 5: The selection of a sample in a study is a crucial process that influences the validity and applicability of the findings. Here are some guidelines and considerations you can take into account when choosing a sample to methodologically better specify sample selection, criteria, etc.
Response 5: We thank the reviewer for suggesting that the sample details be put in the main body of the text as we agree they are crucial to a study’s interpretation. We have included these details in the main body of text now, as opposed to in the supplementary file, and updated them to include more information. Please find this section below:
"Recruitment
Organisation B members (individuals who have given permission to be contacted about research opportunities) received correspondence via email or post with the dates of the support groups, a brief description of the agendas of each support group, and an explanation about the intent to record the meetings for research purposes. Members were made aware that participation was voluntary, and their choice would not affect their membership status. Recruitment closed when each group reached capacity (12 attendees). Participants expressing interest had a short appointment with an Organisation B researcher both to confirm eligibility and to provide the opportunity to discuss the study and ask further questions.
Purposive sampling took place to achieve a sample with people of differing experiences supporting loved ones with different rare dementias.
Participants
11 Organisation B members attended the present support group series. At the groups’ onset, all participants (F=7, M=4) were care partners for a PLWRD (4 posterior cortical atrophy, 3 frontotemporal dementia, 2 primary progressive aphasia, 1 familial Alzheimer’s disease, and 1 dementia with Lewy bodies), although 1 person later became bereaved. The care partners were recruited from across the UK, in a mixture of urban and rural settings. They were mixed in terms of age (43-71 range) and relationship to the PLWRD to ensure the sample represented a diversity of experience to increase the richness of the perspectives captured.
Inclusion criteria required participants to be over 18 years old and to have capacity to understand, retain, assess, and communicate the information required to make the decision to participate in the study. Participants were also required to have a digital device (phone, tablet, or computer) and an internet connection.
The support groups took place virtually via the GoToMeeting (LogMeIn Inc.) video conferencing platform. In accordance with the study procedures outlined in the Participant Information Sheet, the support groups were recorded from the video conferencing platform GoToMeeting with the consent of participants. This platform provides end-to-end security measures preventing active and passive attacks.
Data Pre-processing of Session Files
Collected data were downloaded from the GoToMeeting portals to secure University A servers before being uploaded to the University A Data Safe Haven (certified to the ISO27001 information security standard and conforms to the NHS Digital Data Security and Protection Toolkit).
The recorded data were then transcribed by a third-party transcription service, UK Transcription in accordance with University A guidance."
Comments 6: Figures (3-4) and images some are difficult to visualise and do not present the data, you could revise.
Response 6: We are sorry to hear that the reviewer found these figures (3-4) to not present the data, we felt they were a useful visualisation of the divisions of contribution type in the data and provide a useful measure of how much and when facilitators contributed to the group. We lean towards their inclusion as reviewer 1 made no mention of their inaccessibility but should reviewer 2 feel strongly on this issue we are of course open to reconsidering.
Comments 7: The definitions of rare diseases, as well as those of rare disease carers, to whom they refer, should be improved for better understanding?
Response 7: We hope the reviewer is satisfied with changes made in the introduction when we introduce rare diseases and narrow our focus to rare forms of dementia. We have made efforts to ensure clarity via referring to care partners of people living with dementia (PLWRD) throughout but do see the potential confusion with care partners of people living with rare disease. We have bolded the words dementia and (PLWRD) where the terminology is introduced to hopefully assuage these concerns. The relevant passage introducing these concepts now reads as follows:
"The relative utility of online support groups compared with face-to-face interventions is disputed, but studies have shown both forum-style opportunities (e.g., [4]) and videoconference groups (e.g., [5]) to have positive effects. Findings extend to populations affected by rare diseases such as Cystic Fibrosis [6] and Duchenne’s Muscular Dystrophy [7]. The potential advantages of online intervention are amplified for those affected by rare diseases (conditions affecting fewer than 1 in 2,000 people [8]), where targeted support is seldom locally available.
Another population where this is relevant is people living with rare dementias (PLWRD) – by which we mean atypical (e.g., non-memory-led), young-onset (symptoms/diagnosis before age 65) and directly inherited forms of dementia. Care partners of PLWRD experience the access difficulties outlined above."
Comments 8: The feeling is that they have a lot of information, but in some cases they do not go in depth and sometimes saturating the reader with so much information is better to prioritise and give quality to the data.
Response 8: We hope the reviewer feels we have addressed this comment through the edits we have undertaken to improve readability. This paper is multifaceted and whilst we view that as a strength, if the reviewer strongly feels it should be simplified we can consider this comment further. Examples of how we have simplified include splitting tables, articulating subsections with headings, and further content revision so that our interpretation of the many results is more clearly signposted.
Comments 9: As you comment at the end of your article, the mechanisms by which peer support worked in the groups have been debated at multiple levels. ? In practice, sharing experiences, reflecting on one's own emotional state and responding to others in an open and effective way, in combination with expert facilitation. Theoretically, social comparison theoretical paradigms as useful for understanding the impulses to be in spaces and to respond positively to those spaces and to seek affiliation within them. could better specify these lines of future or improvement.
Response 9: Thank you for this comment, we have edited and further clarified these issues in the future directions and limitations sections and hope that the edits feel suitable. Please see the following section of the limitations and future directions for how we have specified lines of future improvement within Social Comparison Theory.
"We have chosen to evaluate our interpretations against the paradigm of Social Comparison Theory . The findings support the intuition that the support groups offered an opportunity for participants, afflicted by poor experiences of support, to find knowing spaces providing relevant information and experience. When presented with such a space, participants sought comparison and affiliation via sharing with one another and found opportunities for upwards comparison in others and in their expert facilitators. This is pertinent in the context of downward comparison being more prominent in populations facing a health threat [88], and in the context of health behaviour as an important realm of expansion for social comparison research [70]. Future work should continue to consider support groups as an important avenue for social comparison and further integrate the theory into methodology. The current findings also contribute to our understanding of how to develop future tools to allow researchers and facilitators to identify, encourage and understand the different types of social support occurring between participants in online support groups. Such a tool could combine the current quantitative linguistic methods with support codes identified by human qualitative researchers, as well as physiological metrics such as facial emotion recognition."
Round 2
Reviewer 1 Report
Comments and Suggestions for Authors
My reservations about the readability have been addressed. I think you have made it more accessible to a wider audience.
Reviewer 2 Report
Comments and Suggestions for Authors
I want to express my gratitude for the hard work carried out in relation to the suggested modifications to my article. I am very pleased with the responses provided as they have substantially contributed to the improvement of the manuscript.
The clarifications and adjustments made have been extremely helpful for a deeper understanding of the article. I greatly value the effort and attention devoted to this review process.
This research work has been enriched through the collaboration between authors and reviewers. Sometimes, with our experience, reviewers may have different perspectives, but I am pleased to note that we share similar criteria in the pursuit of research excellence.
I sincerely appreciate the commitment and dedication of everyone involved in this process, and I am eager to move this work forward towards its final publication. Congratulations once again.
Best regards